# Dissection of AT-Hook Motif Nuclear-Localized Genes and Their Potential Functions in Peach Growth and Development

**Jianlun Zhao [1,2], Enkai Xu [1] and Qirui Wang [1,*]**

1   College of Landscape Architecture and Art, Henan Agricultural University, Zhengzhou 450046, China; 20201100153@csuft.edu.cn (J.Z.); xek1206@henau.edu.cn (E.X.)
2   College of Life Science and Technology, Central South University of Forestry and Technology, Changsha 410004, China
*   Correspondence: ruwei@henau.edu.cn

**Abstract:** The AT-hook motif nuclear-localized (AHL) family members play key roles in plant biological processes via protein–protein and protein-DNA interactions. Here, 22 non-redundant *PpAHL* genes were identified and analyzed in peach (*Prunus persica*), one of economically important non-timber forestry crops. The maximum-likelihood (ML) tree classified the *PpAHLs* into two clades (Clade-A and Clade-B) with three subfamilies: Type_I, Type_II, and Type_III. Exon–intron analysis exhibited that the *PpAHLs* from Type_I except one (*Prupe.1G530300.1*) lacked introns, and the *PpAHLs* from Type_II and Type_III gradually emerged with intron additions, indicating spatial expression patterns, evolutionarily distinct temporal patterns and, likely, neofunctionalization. Duplication event analysis suggested that *PpAHLs* in peach were mainly expanded through the large-scale duplication events. RNA-seq data showed that *PpAHLs* were induced by drought stress, and two genes (*Prupe.1G530300.1* and *Prupe.1G034400.1*) from Type_I AHLs were induced at all time points, indicating that they might play key roles in the response to drought stress in peach. The tissue-specific expression pattern of *PpAHLs* exhibited their biological functions in the development of these specific tissues. In addition, the transient overexpression of *Prupe.1G530300.1* and *Prupe.1G034400.1* resulted in significant changes in sugar content, suggesting that they may be positive regulators of sugar accumulation in peach fruits. Our study provided novel insights into the roles of *PpAHLs* in plant development, which was helpful for the functional analysis of peach and related woody fruit trees, and for formulating new strategies for further breeding.

**Keywords:** peach; non-timber forestry crops; AHL; expression; duplication; exon–intron

## 1. Introduction

AT-hook motif nuclear-localized (AHL) is a transcription factor, that specifically exists among all land plants, and that is characterized by the inclusion of two conserved domains: one or two AT-hook motifs, and the plants and prokaryotes conserved (PPC) domain [1,2]. The PPC domain played a key role in protein–protein interactions, and the AT-hook motif has been defined as a DNA binding peptide that has been previously described in plants [3,4]. Members of *AHL* family can be clustered into three types: Type_I containing AT-hook 1 and PPC domains, Type_II containing AT-hook 1, AT-hook 2 and PPC domains, and Type_III containing AT-hook 2 and PPC domains [2,5,6].

The *AHL* family members were shown to participate in multiple physiological processes including seed development, lipid biosynthesis, flower development, the regulation of leaf senescence, the regulation of flowering time, the differentiation of vascular tissues, pollen development and fertility, hypocotyl elongation, and responses to biotic and abiotic stresses [4,5,7–17]. For example, *AtAHL18*, as a novel modulator of root system architecture, plays an important role in root growth and development [18]. *AtAHL3* and *AtAHL4* are derived from gene duplication events and both of them can affect the development of

vascular tissue boundaries in Arabidopsis roots [19]. Both *AtAHL20* and *AtAHL22* can regulate flowering time via the repression of *FLOWERING LOCUS* (*FT*) expression [12,20]. *PtrAHL12*, *PtrAHL14*, and *PtrAHL17* were significantly down-regulated at all time points; however, *PtrAHL27*, *PtrAHL29*, and *PtrAHL36* were remarkably up-regulated [5]. The overexpression of *OsAHL1* can enhance tolerance to cold and salt stresses [14].

By scanning the PPC domain and AT-hook motif in the genomes, Li et al. (2022) identified 371 AHLs in 20 land plant genomes [7]. However, the functions of only a few *AHL* genes have been studied in *Arabidopsis*. The fruit tree peach (*Prunus persica*) is one of the economically important non-timber forestry crops, and its small genome makes it a model tree species for fruit tree evolution, genetics, and other biological processes [21]. The productivity of peach trees is constantly limited by abiotic and biotic stresses. Although *AHL* family members may play diverse roles in peach, the current genome-wide analysis of *AHL* remains excluded. The high-quality genome provides us with an excellent opportunity to identify members of the *AHL* gene family and to further analyze their functions in peach. Given the distinct roles of *AHL* family members in various biological processes, the identification of *AHL* genes in peach will help us further understand their functions and lay the groundwork for their future application in the genetic improvement of peach. Similar features of many gene families including *MYB* [22], *CHS* [23], *G3PAT* [24], and *PHD-finger* [25] have helped to identify important target genes to improve fruit quality, stress tolerance, and other important economic traits. The purpose of this study was to systematically analyze the members of the *AHL* family in the model fruit tree peach, so as to increase our understanding of the functional diversity of *AHL* genes during plant development. Our data will provide the basis for improving peach fruit quality and drought tolerance for the *AHL* family.

## 2. Materials and Methods

### 2.1. Data Retrieval

To identify the *AHL* members, we retrieved the latest release of the reference peach genome from the GDR database (https://www.rosaceae.org, accessed on 6 May 2022) [26]. HMMER v3.1 (http://hmmer.org/, accessed on 6 May 2022) was used on the complete peach genome using the hidden Markov model (HMM) of PPC/DUF296 (PF03479) downloaded from Pfam (https://pfam.xfam.org/, accessed on 10 May 2022) [27,28]. The Conserved Domains Database (https://www.ncbi.nlm.nih.gov/Structure/cdd/wrpsb.cgi, accessed on 10 May 2022) was used to confirm the presence of characteristic domains in the obtained candidate sequences using an e-value of $10^{-5}$ [29]. Sequences were considered putative AHLs if they contained both an AT-hook motif and a PPC domain. The putative sequences obtained were again confirmed by searching the NCBI database (https://www.ncbi.nlm.nih.gov/, accessed on 22 May 2023) with an e-value of $10^{-5}$.

### 2.2. Genes Structure

The MEME server (https://meme-suite.org/meme/, accessed on 18 June 2022) [30] and GFF3 file were used to predict the motifs of AHL proteins and gene structure in *AHL* genes, respectively. The IMEter server (https://korflab.github.io/, accessed on 18 June 2022) was implemented to analyze the effect of introns on gene expression enhancement [31]. The effect of introns on gene expression was displayed using an IMEter score of 0 to 100. If the score was greater than 20, it was considered that the effect of introns on the increase in expression was very high [31,32].

### 2.3. Phylogenetic Analysis

A maximum likelihood (ML) tree was constructed to infer evolutionary history using IQ-tree v1.6.12 (http://www.iqtree.org/release/v1.6.12, accessed on 15 June 2023) [33]. All AHL amino acid sequences were aligned using MAFFT v7 in PhyloSuite (https://dongzhang0725.github.io/, accessed on 15 June 2023) [34]. The 10,000 bootstrap replicates and 1000 SH-aLRT random addition replicates were applied to prepare this consensus

to represent the phylogenetic relationships [7]. The large-scale duplication events were obtained via MicroSyn v1 [35].

### 2.4. Gene Expression Analysis

The RNA-seq data for different tissues or fruit development obtained from NCBI (https://www.ncbi.nlm.nih.gov/, accessed on 15 June 2023) with accession numbers of PRJNA694007 and PRJNA694331 were used for the expression of *PpAHLs*. These reads were aligned to the peach genome (Prunus_persica_v2.0.a1_scaffolds.gff3) with HISAT2 v2.0.4 (http://daehwankimlab.github.io/hisat2/, accessed on 15 June 2023) followed by the estimation of fragments per kilobase million (FPKM) with StringTie v2.2.1 (https://github.com/gpertea/stringtie, accessed on 15 June 2023) using default parameters [36,37], as described by Ding et al. (2020) [38]. Finally, we used the log2-transformed values of FPKM to prepare heatmaps.

### 2.5. qRT-PCR Analysis

We collected peach cultivar 'Xia Huanjin' fruit samples on 20 June (125 days) corresponding to the ripening stage in 2022 after flowering (DAF) from 20-year- old pear trees grown on a farm in Hefei, Anhui, China. The peach fruits were sprayed with 25% polyethylene glycerol-6000 (PEG) to induce drought stress. The sampled fruits were collected at 0 and 12 h (h) after treatment and immediately frozen in liquid nitrogen for later use. Total RNA Rapid Extraction Kit (Magen, Guangzhou, China) was used to extract the total RNA for all obtained samples. PrimeScript™ RT Reagent Kit with gDNA Eraser (Takara, Dalian, China) was used to synthesize the first-strand cDNA. SYBR® Premix Ex Taq™ II (Takara Bio Inc., Kusatsu City, Japan) was used to carry out the qRT-PCR, based on the manufacturer's instructions. The PCR program consisted of 40 cycles at 95 °C for 10 s and 60 °C for 30 s, separated by a 95 °C cycle for 1 min. The *PpTEF2* was used as an internal control for normalization [39]. Primers (Table S1) were designed using a primer-designing tool (https://www.ncbi.nlm.nih.gov/tools/primer-blast/, accessed on 15 May 2023).

### 2.6. Subcellular Localization Analysis of Prupe.1G530300.1 and Prupe.1G034400.1

The full-length coding sequences (CDS) of *Prupe.1G530300.1* and *Prupe.1G034400.1* were cloned from peach and inserted into the pCambia1304 vector (Clontech, Beijing, China), which contains a CaMV 35S promoter and *GFP* gene. The resulting constructs produced fusion proteins of Prupe.1G530300.1-GFP and Prupe.1G034400.1-GFP. The Prupe.1G530300.1-GFP and Prupe.1G034400.1-GFP constructs were electroporated into *Agrobacterium* tumefaciens strain EHA105 using Gene Pulser Xcell (BIO-RAD, Heracles, CA, USA). The suspensions were injected into the leaves of *Nicotiana tabacum* using the infiltration method, and the expressed Prupe.1G530300.1-GFP and Prupe.1G034400.1-GFP genes were observed using confocal laser scanning microscopy (Carl Zeiss LSM710, Oberkochen, Germany).

### 2.7. Transient Overexpression in Peach Fruits

The full-length coding sequences of *Prupe.1G530300.1* and *Prupe.1G034400.1* were introduced into pSAK277 and then transformed into the *A. tumefaciens* strain GV3101. The transformed *A. tumefaciens* was grown in a culture until the OD 600 nm ranged from 1.5 to 2.0 at 28 °C. Centrifugation of *A. tumefaciens* at $1000 \times g$ for 5 min was performed to collect it. The precipitate was resuspended in the infiltration buffer, which was made up of 1 M NaOH, 10 mM 2-(N-morpholino) ethanesulphonic acid (MES), 150 M acetosyringone, and 10 mM MgCl$_2$. *A. tumefaciens* was utilized to contaminate 'Xia Huanjin' peach fruits at the ripening stage after being diluted to an OD 600 nm value of roughly 0.4. One side of the fruit received injections of *A. tumefaciens* cultures containing the candidate gene, while the other side received injections of *A. tumefaciens* cultures containing the empty vector as a control. Each treatment underwent a minimum of three biological replications, and samples were taken five days following infiltration.

### 2.8. Measurement of Fruit Sugar

High-performance liquid chromatography (HPLC) was used to measure the type and quantity of sugar. An Agilent 1260 Infinity HPLC system (Milford, MA, USA) fitted with a refractive index detector (Shodex RI-101; Shodex Munich, Germany) was used to evaluate the sugar content of fruit samples. To conduct separation, a Transgenomic COREGET-87C column (7.8 mm × 300 mm × 10 m) with a guard column (Transgenomic CARB Sep Coregel 87C) was utilized. A Dionex TCC-100 thermostated column compartment kept the column temperature at 85 °C. With degassed, distilled, and deionized water, the flow rate at the mobile phases was kept at 0.6 mL/min. The fresh weight (FW) method was used to express sugar concentrations. Sucrose, fructose, and glucose amounts were used to determine the total amount of sugar.

### 3. Results and Discussion

### 3.1. Systematic Identification Reveals PpAHL Diversification in Peach

A total of 22 PpAHL proteins were identified in the peach genome that contained both an AT-hook motif and a PPC domain (Table 1). The number of *PpAHLs* in peach is lower than that in Arabidopsis (29) [40], poplar (37) [5], maize (37) [41], soybean (63) [17], and carrot (47) [42], and higher than those in castor (16) [2], tung tree (19) [2], and grape (19) [6], suggesting *AHLs* in different plant genomes have different degrees of expansion. To determine the phylogenetic relationship between *AHLs*, we constructed a ML tree using the full-length proteins which divided these *AHLs* into two clades, Clade-A and Clade-B according to the topology of the tree and bootstrap values. Clade-A contains 12 *PpAHLs* and Clade-B consists of 10 *PpAHLs* (Figure 1). These results are very similar to those reported previously [5,40]. For example, Li et al. (2022) analyzed the *AHLs* of 20 plants and also divided them into two clades [2]. We further performed multiple-sequence alignment of *PpAHLs* to examine two characteristic domains, AT-hook and PPC (Figure 2), and found that these *PpAHL* sequences could be further classified into Type_I, Type_II, and Type_III. Of the 22 *PpAHLs*, 12 (54.5%) belong to Type_I, 4 (18.2%) correspond to Type_II, and 6 (27.3%) are Type_III. The *PpAHLs* from Type_I belong to Clade-A while the *PpAHLs* from Type_II and Type_III belong to Clade-B. The proportions of three different types of *PpAHLs* in two different clades in the peach genome are basically similar to those in other plants [2,6,40]. For example, although Clade-B contains members of Type_II and Type_III, which has fewer *PpAHL* members, Clade-A (Type_I) contains the most *PpAHL* members. These results imply that the evolutionary diversity of *AHLs* is well-conserved in both monocot and eudicot plants.

**Table 1.** The detailed information of PpAHLs in peach.

| Gene ID | Chr | 5′ End | 3′ End | Strands | Types |
|---|---|---|---|---|---|
| *Prupe.1G530300.1* | Pp01 | 43,345,339 | 43,347,157 | − | Type_I |
| *Prupe.4G266900.1* | Pp04 | 20,020,558 | 20,021,945 | − | Type_I |
| *Prupe.5G004600.1* | Pp05 | 578,564 | 581,015 | − | Type_I |
| *Prupe.8G182400.1* | Pp08 | 18,209,645 | 18,211,172 | + | Type_I |
| *Prupe.5G037400.1* | Pp05 | 4,194,916 | 4,195,767 | − | Type_I |
| *Prupe.2G108200.1* | Pp02 | 16,651,246 | 16,652,628 | + | Type_I |
| *Prupe.1G034400.1* | Pp01 | 2,401,099 | 2,401,956 | − | Type_I |
| *Prupe.6G348200.1* | Pp06 | 29,755,598 | 29,756,515 | − | Type_I |
| *Prupe.2G239000.1* | Pp02 | 25,943,216 | 25,946,736 | − | Type_I |
| *Prupe.3G173700.1* | Pp03 | 19,109,639 | 19,110,439 | − | Type_I |
| *Prupe.2G147000.1* | Pp02 | 20,331,029 | 20,332,901 | + | Type_I |
| *Prupe.5G082200.1* | Pp05 | 9,555,987 | 9,557,995 | + | Type_I |
| *Prupe.5G098800.1* | Pp05 | 10,701,796 | 10,707,244 | − | Type_II |
| *Prupe.2G167000.1* | Pp02 | 21,560,234 | 21,565,660 | − | Type_II |
| *Prupe.5G037000.1* | Pp05 | 4,177,031 | 4,182,789 | + | Type_II |
| *Prupe.5G005600.1* | Pp05 | 668,431 | 673,030 | + | Type_II |
| *Prupe.6G347600.1* | Pp06 | 29,734,568 | 29,739,564 | + | Type_III |

**Table 1.** *Cont.*

| Gene ID | Chr | 5′ End | 3′ End | Strands | Types |
|---|---|---|---|---|---|
| *Prupe.5G081500.1* | Pp05 | 9,472,759 | 9,478,136 | − | Type_III |
| *Prupe.2G282300.1* | Pp02 | 28,057,573 | 28,061,864 | − | Type_III |
| *Prupe.2G146200.1* | Pp02 | 20,256,616 | 20,262,297 | − | Type_III |
| *Prupe.6G117900.1* | Pp06 | 8,644,708 | 8,647,802 | + | Type_III |
| *Prupe.7G119600.1* | Pp07 | 14,344,867 | 14,348,069 | − | Type_III |

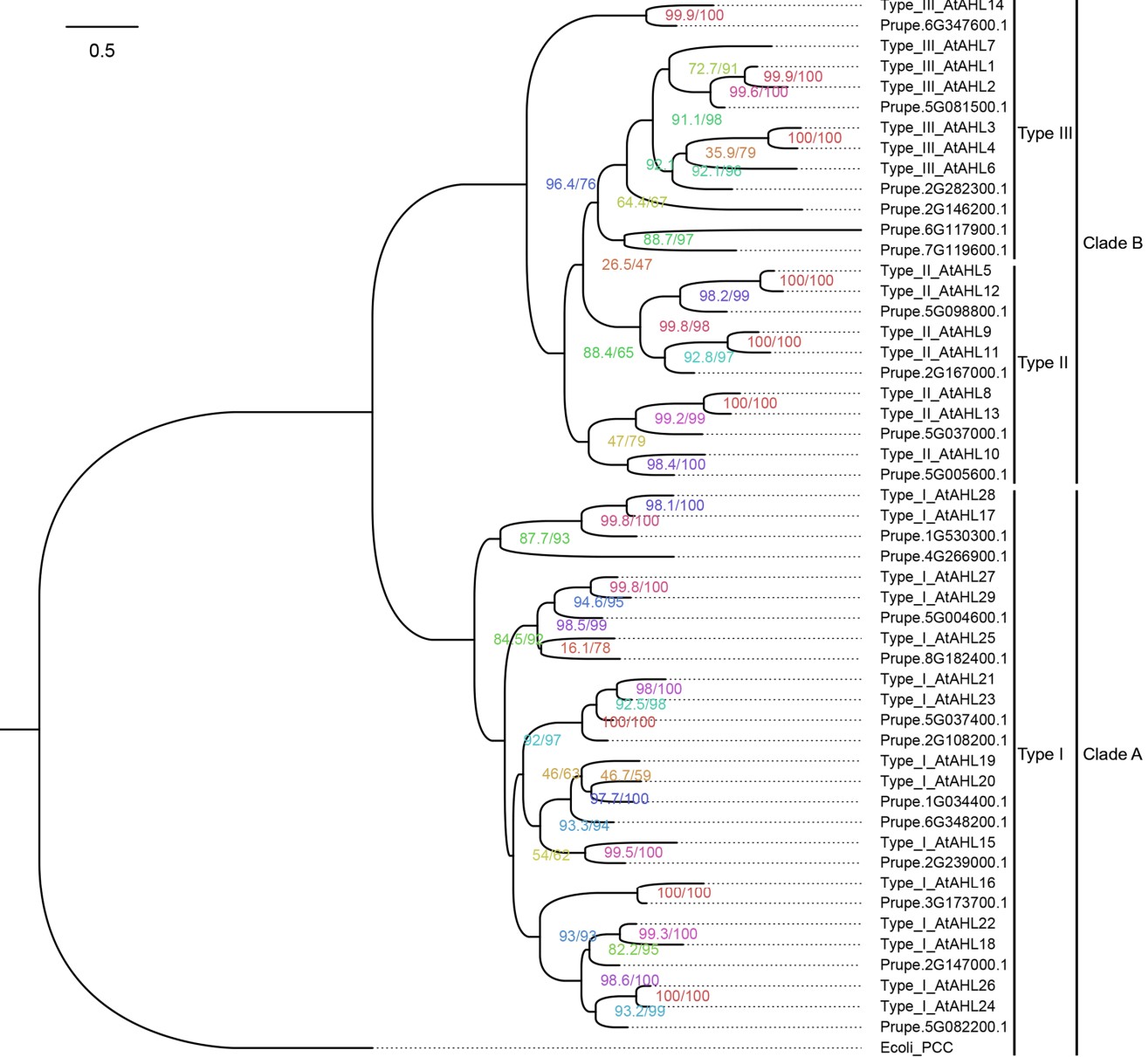

**Figure 1.** Phylogenetic analysis of the peach *PpAHLs*. This tree was generated using IQ-tree with the maximum likelihood (ML) method. The 10,000 bootstrap replicates and 1000 SH-aLRT random addition replicates were applied to prepare this consensus to represent the phylogenetic relationships.

The AT-hook motifs from Type_I and Type_II differ according to the conserved sequence flanking the Arg-Gly-Arg core, particularly the carboxy-terminal sequence (Figures 2 and S1–S3). The conserved sequence in AT-hook motifs of Type_I PpAHLs is Gly-Ser-Lys-Asn-Lys on the C-terminus (Figure 2), whereas this sequence in AT-hook motifs of Type_II PpAHLs is Arg-Lys-Tyr-X. Additionally, one additional AT-hook motif was also

found in Type_II PpAHLs. The PPC domain in Type_I PpAHLs lacks the less-conserved sequence Leu-Arg-Ser-His, and this domain in Type_II and Type_III PpAHLs lacks the less-conserved sequence Phe-Thr-Pro-His (Figures S1–S3). This structure of PpAHLs in peach is consistent with the evolutionary relationship described above, in which the PPC domain is already present in ancient PpAHLs and the addition of one or more AT-hook motif(s) in flowering plants diversifies AHLs [7,43].

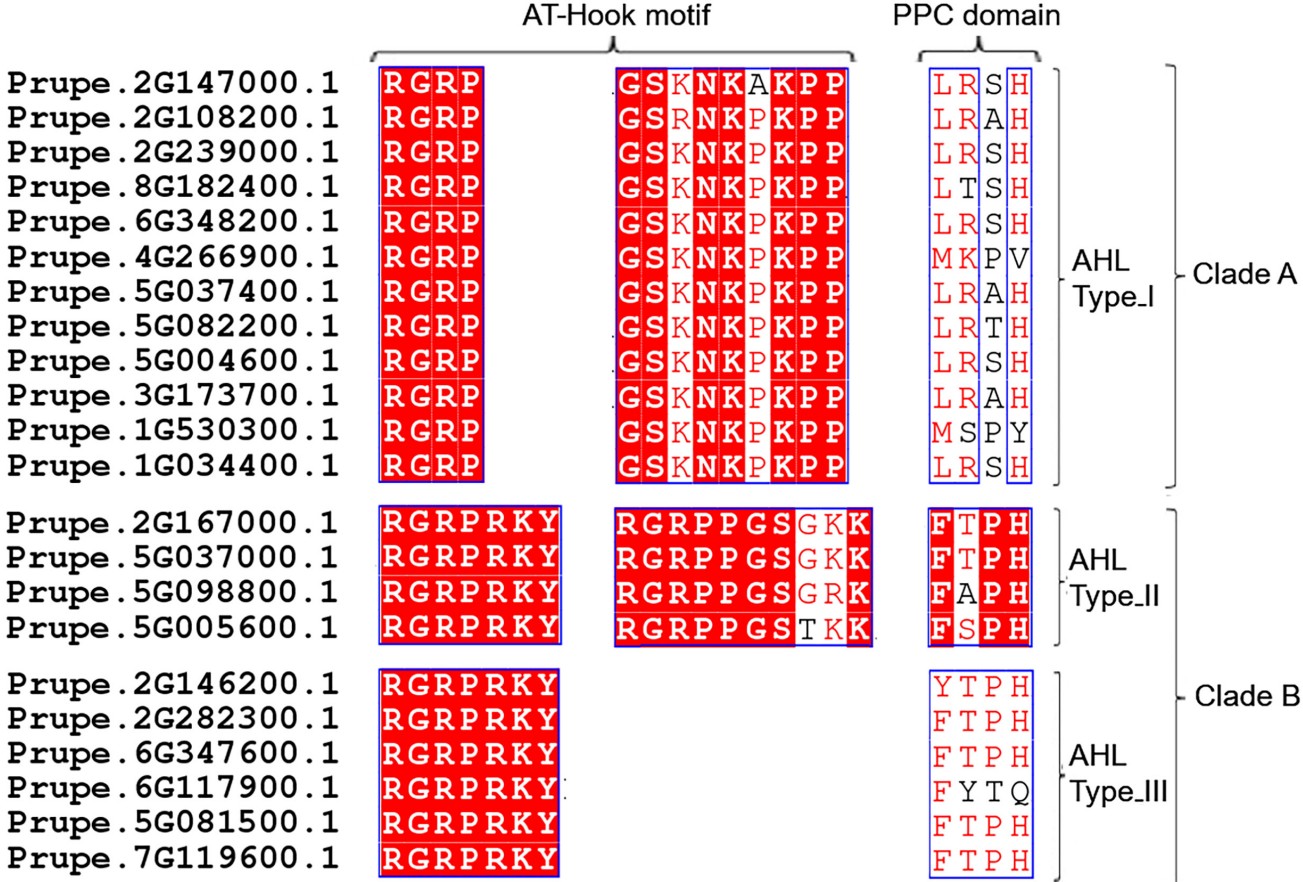

**Figure 2.** The characteristic sequences in two functional units of the PpAHLs in peach. The conserved sequences were obtained from multiple-sequence alignment analysis performed using MAFFT v7 software.

### 3.2. Intron Gain or Loss during Diversification of PpAHLs Indicates Intron-Mediated Modification of Expression

The exon–intron analysis exhibited that the number of introns and exons in the peach *PpAHLs* was large and indicated diversity. In our study, the *PpAHLs* from Type_I except one (*Prupe.1G530300.1*) lacked introns, while *PpAHLs* from Type_II and Type_III contained four introns except one (*Prupe.6G347600.1*) which had five introns (Figure 3). Prokaryotic genes do not contain introns, and eukaryotic genes acquired introns after or during eukaryotic–prokaryotic differentiation [44–46]. Thus, we considered Type_I *PpAHLs* to be the earliest ancestral *AHLs*, which then gradually evolved into Type_II and Type_III *PpAHLs* with the evolution of *AHLs*. The evolution of introns is related to alternative splicing and the resulting nonsense-mediated decay of mRNA transport [47], enhancement of gene expression [48], and functional diversity of coding proteins [49–51]. *AHL* genes have diversified in this form and their function has evolved following subsequent differentiation and duplication [7,41].

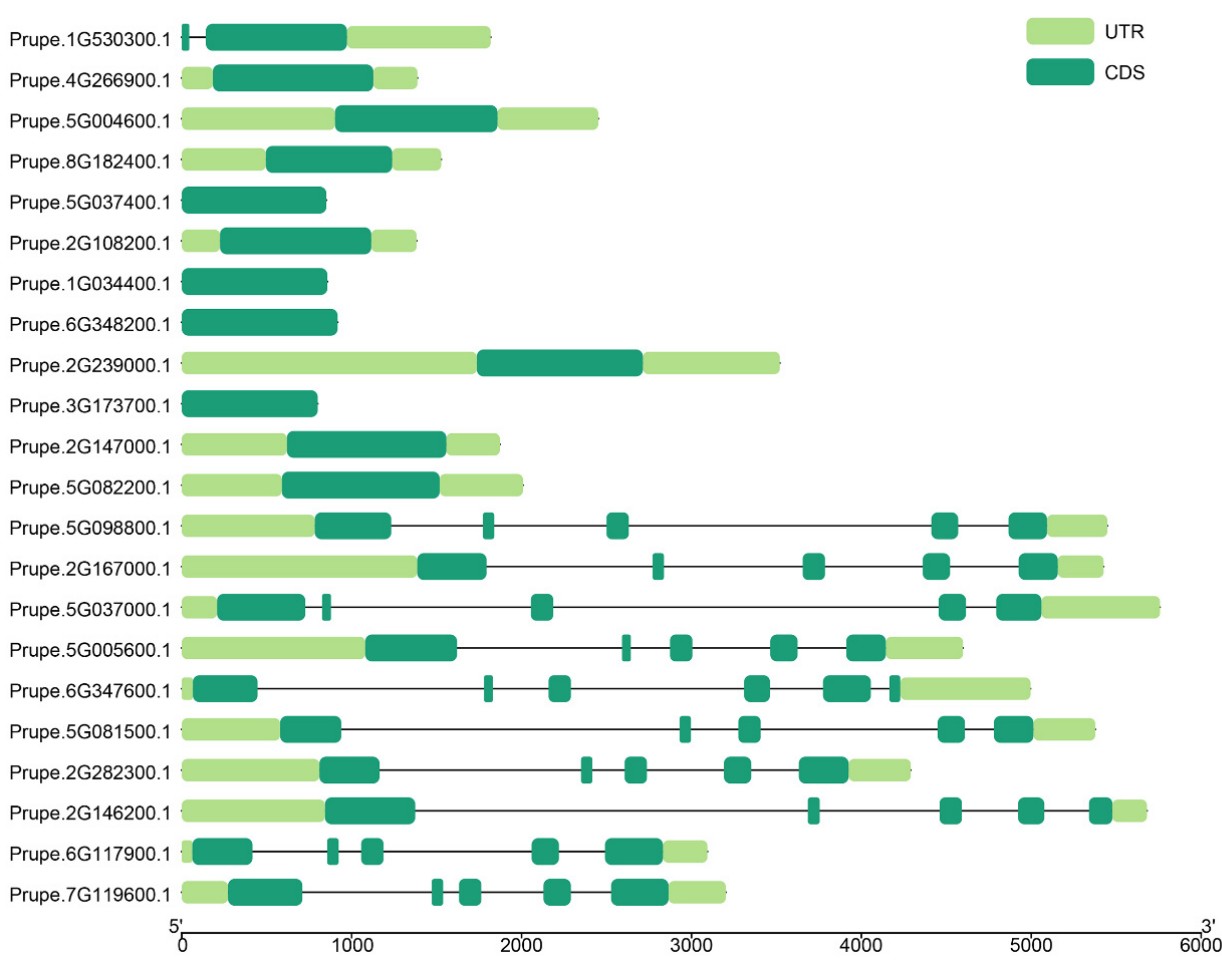

**Figure 3.** Exon–intron structure of *PpAHLs* in peach. The untranslated regions (UTRs), introns, and exons are indicated by green rectangles, thin lines, and blue rectangles, respectively.

Introns in various organisms, including fungi, animals, and plants, can increase the expression of their genes [31,48,52]. To determine the biological roles of introns in *PpAHLs*, the capabilities of these introns were examined to increase the expression of the associated genes. The first introns of one Type_II *PpAHL* gene (*Prupe.5G005600.1*) and two Type_III *PpAHL* genes (*Prupe.5G081500.1* and *Prupe.2G282300.1*) were predicted to enhance the transcript abundance of these genes. This result was consistent with that of the subsequent expression analyses; that is, these three genes were highly expressed in the analyzed tissues. Taken together, the increase in introns in Type_II and Type_III *AHL* genes suggests that they play important roles in processes associated with enhanced biological complexity, as reported in maize by Bishop et al. (2020). Remarkably, we found that one intron with a Type_I *PpAHL* (*Prupe.1G530300.1*) seems to be the evolutionary link between Clade-A (Type_I) and Clade-B (Type_II and Type_III), which was supported by the phylogenetic analysis. The intron added in *Prupe.1G530300.1* could enhance its expression abundance and regulation, indicating that the origin of this gene might be related to the acquisition of corresponding different functions, as reported in maize by Bishop et al. (2020). This neofunctionalization might facilitate the adaptation of modern plants to changing environments, providing a source of motivation for the further expansion of intron-carrying *AHL* genes.

*3.3. Gene Duplication Analysis Suggests the Large-Scale Duplications Contribute to the Increase in PpAHLs in Peach*

Gene duplication events may involve the expansion of gene members in peach [21,22,25]. Here, we constructed the gene duplication analysis of *PpAHLs* to reveal the evolutionary

process. The *PpAHLs* were used as anchor sites to detect duplication events in the regions where they reside. The flanking genes containing the anchor site *PpAHL* were aligned, and if there were three or more pairs that were microsynteny, they were considered to be derived from a large-scale duplication event [53,54].

Here, we detected that nine *PpAHL* genes likely originated from independent duplication events as they did not contain conserved microsynteny (Figure 4). In total, 10 *PpAHLs* pairs were identified whose flanking genes contained at least three gene pairs (Figure 4 and Table S2). Among them, three conserved genes flanking five pairs were identified, including *Prupe.2G146200.1/Prupe.2G147000.1*, *Prupe.2G146200.1/Prupe.2G282300.1*, *Prupe.2G239000.1/Prupe.4G266900.1*, *Prupe.2G282300.1/Prupe.5G082200.1*, and *Prupe.5G004600.1/ Prupe.8G182400.1*. Five other pairs of *PpAHLs* (*Prupe.2G146200.1/Prupe.5G081500.1*, *Prupe.2G146200.1/Prupe.5G082200.1*, *Prupe.6G117900.1/Prupe.7G119600.1*, *Prupe.2G167000.1/ Prupe.5G098800.1*, and *Prupe.5G081500.1/Prupe.5G082200.1*) were found to possess more than three pairs of conserved flanking genes. Taken together, our study suggested that the large-scale duplication event contributed to the expansion of *PpAHLs* in peach. Indeed, many studies have confirmed that this duplication plays key a role in the expansion of *AHLs* in plants [2,6,41].

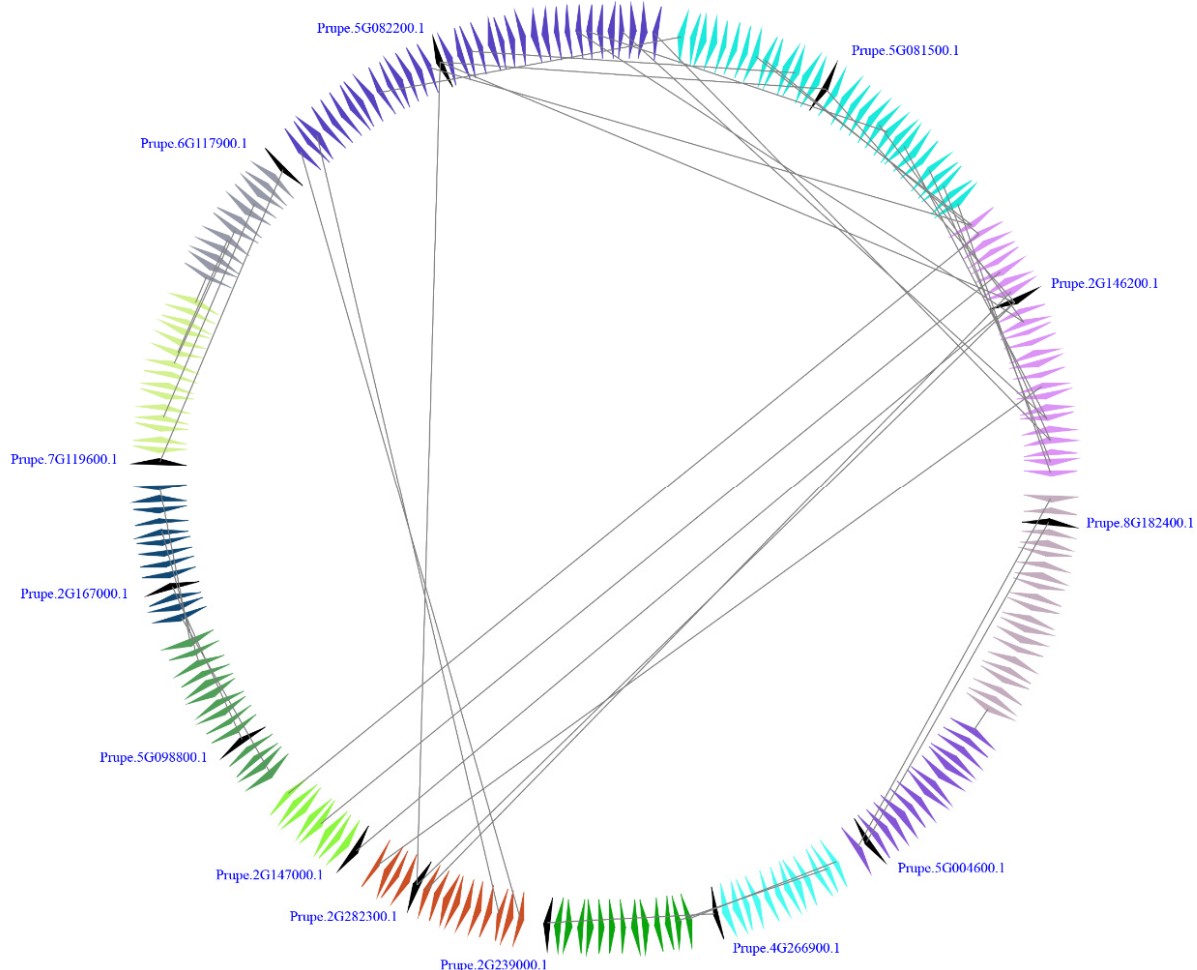

**Figure 4.** The large-scale duplication event analysis of *PpAHLs* in peach. The triangle indicates both the gene's orientation and *PpAHL* flanking genes. The homologs are connected by a series of gray lines.

### 3.4. Expression Patterns of PpAHLs in Response to Drought Stress

Water is necessary to maintain normal plant physiological processes such as reproduction, development, and growth [55–57]. Woody plants have developed strategies to

overcome and survive drought, such as reprogramming gene expression [58]. Therefore, we believe that it is very necessary to determine the regulatory pathways and key factors of peach response to drought stress. To determine the stress responses involving the *PpAHLs*, the RNA-seq data were used to analyze their expressions under drought stress. As shown in Figure 5, we found that a wide variety of *PpAHL* expression patterns were exhibited in our study. Four genes (*Prupe.4G266900.1*, *Prupe.5G037400.1*, *Prupe.3G173700.1*, and *Prupe.6G117900.1*) were not detected in response to drought stress, implying that these genes might be involved in other biological processes. A total of eight *PpAHLs* were down-regulated under drought stress, while two genes (*Prupe.1G530300.1* and *Prupe.1G034400.1*) from Type_I AHLs were induced at all time points, indicating these genes might play important roles in the response to drought stress in peach. To confirm these results, a qRT-PCR analysis was carried out (Figure S4). As expected, two genes (*Prupe.1G530300.1* and *Prupe.1G034400.1*) exhibited expression patterns similar to those in the RNA-Seq data. Indeed, Wang et al. (2021) also carried out an expression analysis of *PtrAHLs*, and suggested that *PtrAHL8*, *PtrAHL9*, *PtrAHL16*, *PtrAHL20*, *PtrAHL23*, *PtrAHL27*, *PtrAHL29*, *PtrAHL34*, and *PtrAHL36* were induced under drought stress in both leaves and roots [5]. Additionally, *Prupe.1G530300.1* and *Prupe.1G034400.1* were the homolog of *OsAHL1* (*LOC_Os11g05160*) that could enhance drought tolerance by binding its target *AP2-EREBP*, *Rab16b*, *OsRNS4*, *OsCDPK7*, and *HSP101* [14].

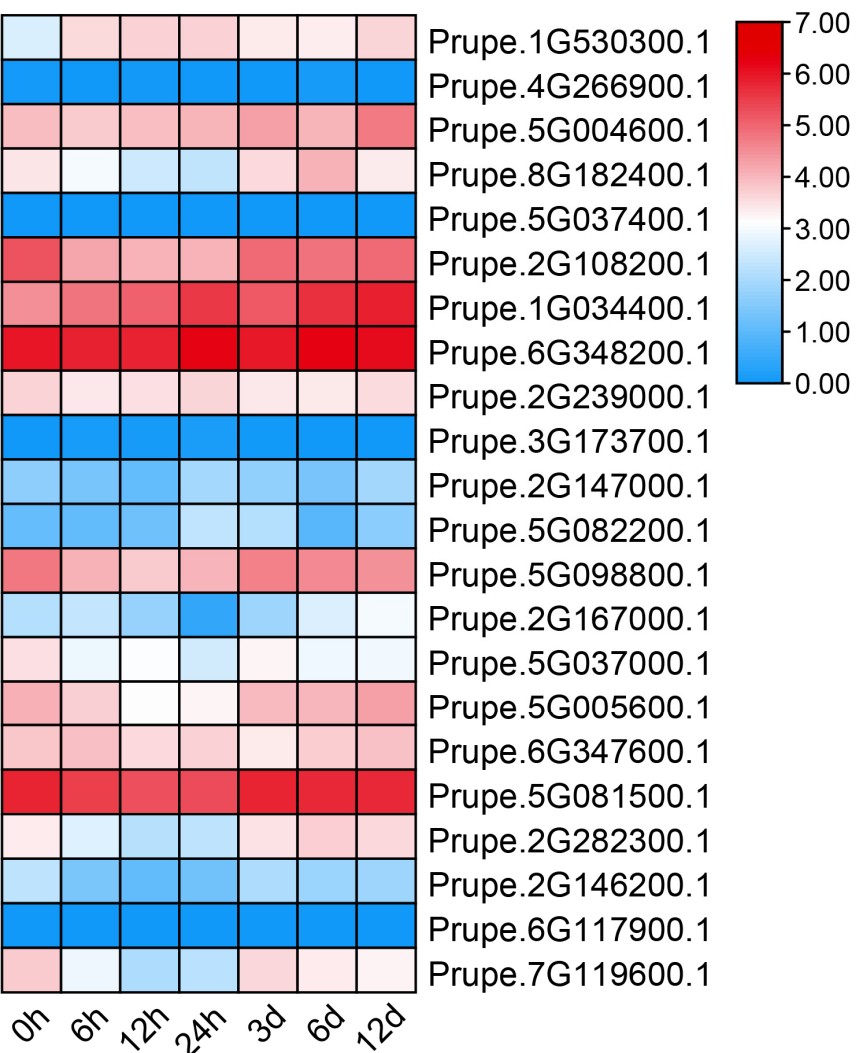

**Figure 5.** *PpAHL* differential expression under drought stress assessed through RNA-seq data. The color scale indicates the values of a log2 fold change; blue represents a low level and red suggests a high level of transcript abundances.

### 3.5. Expression Profile in Different Tissues or during Development Provides Important Insights into Biological Functions of PpAHLs in Peach

The *AHL* family members play essential roles in plant growth, floral transition and during fruit development [7,8,16]. To explore the potential biological functions of *PpAHLs* during development, we determined the expression profile of 22 *PpAHLs* in six tissues, including phloem, roots, seeds, flower, fruits, and leaves. In our study, six *PpAHLs* (*Prupe.4G266900.1*, *Prupe.5G037400.1*, *Prupe.3G173700.1*, *Prupe.2G146200.1*, *Prupe.6G117900.1*, and *Prupe.7G119600.1*) were not detected in any tissues, suggesting that they might be pseudogenes or genes whose expression was induced by environmental stresses (Figure S5). Most *PpAHLs* from Type_I AHLs (such as *Prupe.5G004600.1*, *Prupe.2G108200.1*, *Prupe.1G034400.1*, *Prupe.6G348200.1*, *Prupe.2G147000.1* and *Prupe.5G082200.1*) exhibited the highest transcript accumulation in roots, but extremely low transcript accumulation in other tissues, such as leaves, flower, and fruits. The *PpAHLs* from Type_II and Type_III AHLs (such as *Prupe.5G098800.1*, *Prupe.2G167000.1*, *Prupe.5G037000.1*, *Prupe.5G005600.1*, and *Prupe.5G081500.1*) exhibited extensive expression activity in almost all tissues, suggesting that these genes might be indispensable for plant growth and development. Interestingly, three genes, *Prupe.5G005600.1* from Type_II, and *Prupe.5G081500.1* and *Prupe.2G282300.1* from Type_III, inserted via the introns in peach exhibited higher transcript accumulation in almost tissues than that in other *PpAHLs*, which is consistent with that in *Zm00001d051861* and *Zm00001d018515*, which were constitutively expressed in all organs [41].

### 3.6. Subcellular Localization of Prupe.1G530300.1 and Prupe.1G034400.1

The nuclear localization of transcription factors (TFs) is crucial for their regulatory functions. Several factors impact the nuclear entry process, such as environmental stress, the cell cycle, and the developmental stage. Although 40–60 kD proteins can pass through nuclear pores via diffusion, the movement of a protein through a nuclear pore is an active process that requires one or more nuclear localization signals. To investigate the subcellular localization of *Prupe.1G530300.1* and *Prupe.1G034400.1*, expression vectors for Prupe.1G530300.1-GFP and Prupe.1G034400.1-GFP were constructed and introduced into *N. tabacum*. As depicted in Figure 6A, the green fluorescence signals emitted from the expressed fusion genes of Prupe.1G530300.1-GFP and Prupe.1G034400.1-GFP were selectively distributed within the nuclei.

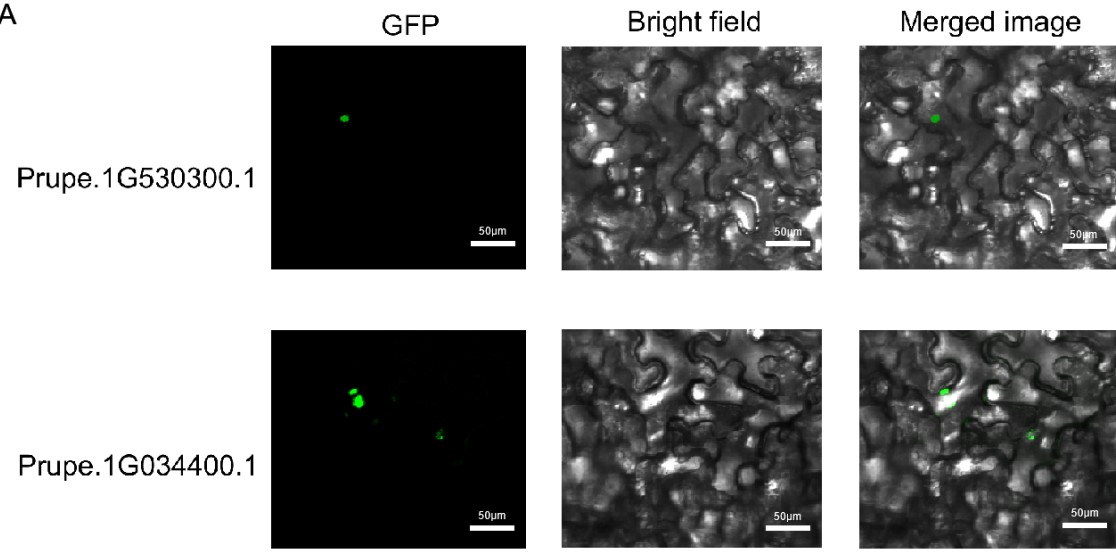

**Figure 6.** *Cont.*

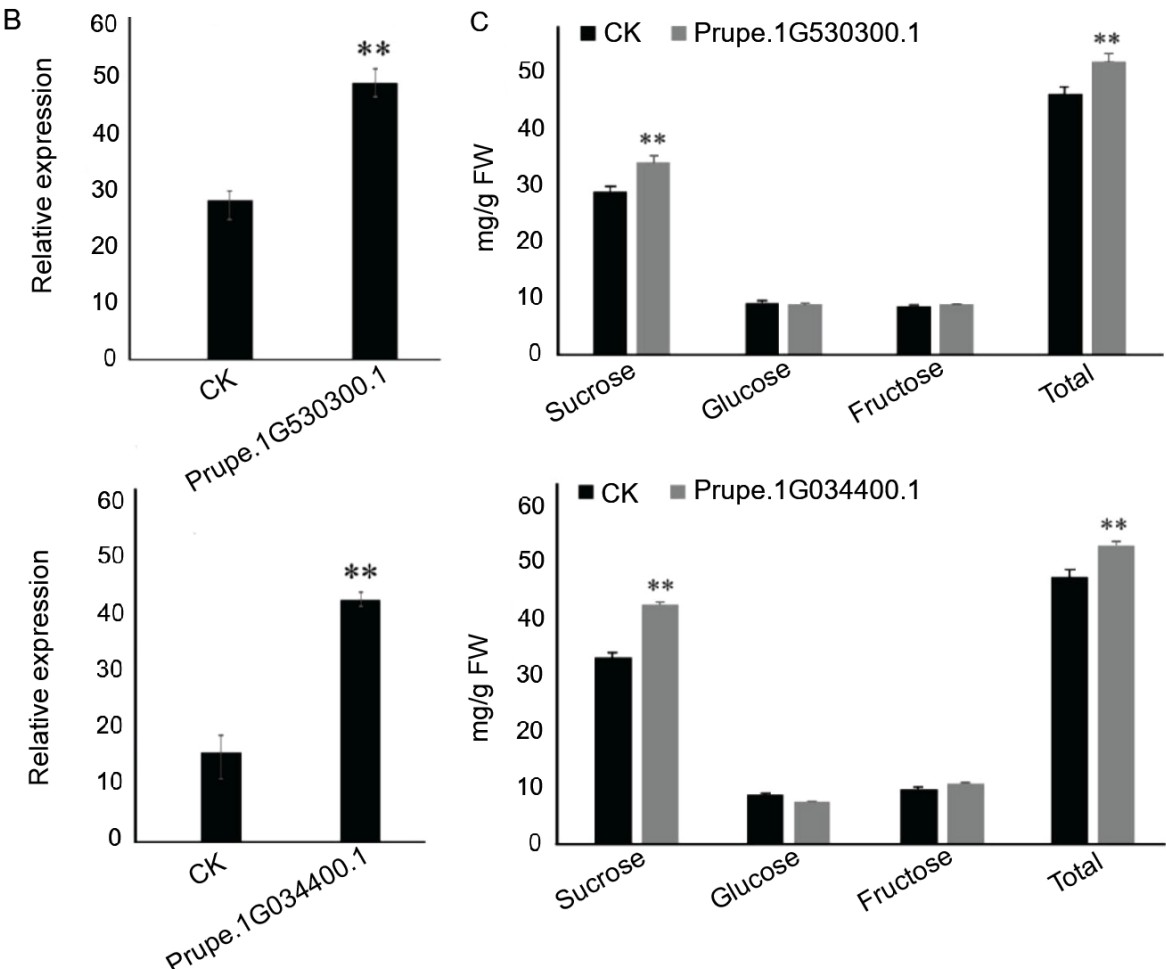

**Figure 6.** Functional analysis of *Prupe.1G530300.1* and *Prupe.1G034400.1*. (**A**) Analysis of subcellular localization of *AHL* genes; (**B**) expression analysis of *Prupe.1G530300.1* and *Prupe.1G034400.1* in peach fruits; (**C**) functional analysis of *Prupe.1G530300.1* and *Prupe.1G034400.1* via their transient overexpression in peach fruits. The bars on the sugar content chart represent the standard error (SE) of the measurements, ** $p < 0.01$.

### 3.7. Transient Overexpression of Prupe.1G530300.1 and Prupe.1G034400.1 Had the Opposite Effect on Sugar Accumulation in Peach Fruits

*Prupe.1G530300.1* and *Prupe.1G034400.1* were discovered to function as potential positive regulators of sugar accumulation [59]. We carried out functional analyses using their transient overexpression in peach fruits to validate this discovery. Gene expression levels and sugar content in the fleshy tissue surrounding the infiltration sites were assessed five days after transformation. As a result, the expression level of *Prupe.1G530300.1* and *Prupe.1G034400.1* in the flesh tissues infiltrated with *Prupe.1G530300.1* and *Prupe.1G034400.1* was significantly higher than that in the flesh tissues infiltrated with the empty vector (EV) (Figure 6B). The contents of sucrose and total sugar in the flesh tissues infiltrated by *Prupe.1G530300.1* and *Prupe.1G034400.1* were significantly higher than those in the flesh tissues infiltrated by EV (Figure 6C), which is consistent with the prior finding of a positive correlation between the expression of *Prupe.1G530300.1* and *Prupe.1G034400.1* and the content of either total sugar or sucrose. Taken together, our data suggest that *Prupe.1G530300.1* and *Prupe.1G034400.1* are both involved in the regulation of sugar accumulation in peach fruits.

## 4. Conclusions

The importance of *AHLs* in economic traits related to plant growth and development and drought tolerance is evident from studies in rice and Arabidopsis. The significant lack of knowledge about the potential roles of *AHLs* in woody economic plants indicates the importance of the identification and functional characterization of the *AHL* family. Crop yield is closely related to quality and plant resistance to environmental stress; therefore, the identification and functional characterization of *AHLs* in peach, one of the world's most important economic fruit trees, has great implications for the improvement of woody economic plants. Here, we identified 22 *PpAHLs* and further performed a comprehensive analysis of these genes in peach. According to the RNA-seq data, we used a systematic approach to further obtain the potential biological functions of *PpAHLs* in response to drought stress. The overexpression of *Prupe.1G530300.1* and *Prupe.1G034400.1* resulted in significant changes in sugar content, suggesting that they may be positive regulators of sugar accumulation in peach fruits. Our findings help to identify candidate genes involved in plant growth and development or in response to drought stress, which is of great significance for a comprehensive understanding of the regulatory roles of *AHL* family members.

**Supplementary Materials:** The following supporting information can be downloaded at: https://www.mdpi.com/article/10.3390/f14071404/s1, Table S1. Primers in this study. Table S2. Paralogous data of PpAHLs in peach. Figure S1. Multiple sequence alignment analysis of Type_I PpAHLs in peach. Figure S2. Multiple sequence alignment analysis of Type_II PpAHLs in peach. Figure S3. Multiple sequence alignment analysis of Type_III PpAHLs in peach. Figure S4. qRT-PCR verification of two genes (*Prupe.1G530300.1* and *Prupe.1G034400.1*) in response to drought stress in peach fruits. The standard error bar indicates three biological replicates. Figure S5. The tissue-specific expression patterns of *PpAHLs* in different tissues. The color scale indicated the values of log2 fold change, blue represented low level and red suggested high level of transcript abundances.

**Author Contributions:** J.Z. and Q.W. planned and designed the experiments. J.Z., E.X. and Q.W. performed the experiments and the sequence analysis. J.Z. and E.X. wrote the manuscript. Q.W. revised the manuscript. All authors have read and agreed to the published version of the manuscript.

**Funding:** This work was supported by the Major Science and Technology Special Projects in Henan Province (201300111400).

**Data Availability Statement:** All data can be found online in the main text and supporting information materials. The RNA-seq data can be found in the Sequence Read Archive (SRA) database with accession numbers PRJNA694007 and PRJNA694331.

**Conflicts of Interest:** The authors declare that the research was conducted in the absence of any commercial or financial relationships that could be construed as potential conflicts of interest.

## Abbreviations

AT-hook motif nuclear-localized: AHL; maximum likelihood: ML; prokaryotes conserved: PPC; fragments per kilobase million: FPKM; hidden Markov model: HMM.

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
