# Peer review of "Dissection of AT-Hook Motif Nuclear-Localized Genes and Their Potential Functions in Peach Growth and Development"

_forests, doi:10.3390/f14071404_

Round 1

Reviewer 1 Report

Results of the study will have valuable contribution for new strategies for further breeding, and systematic of peach (Prunus persica).

1. What is the main question addressed by the research? The study addressed to management strategies of breeding, and systematical analyze of peach (Prunus persica).

2. Do you consider the topic original or relevant in the field? Does it address a specific gap in the field? The topic is original and addresses a specific gap in the field.

3. What does it add to the subject area compared with other published material? Results of the study will have valuable contributions for the addresses. The study can be accepted as one of the first paper in the field of the species. Results of the study can be guide for future studies.

4. What specific improvements should the authors consider regarding the methodology? What further controls should be considered? It is detail and sufficient.

5. Are the conclusions consistent with the evidence and arguments presented and do they address the main question posed? They are extracted from results of the paper.

6. Are the references appropriate? References are appropriate

7. Please include any additional comments on the tables and figures. All table and figures are related to the paper. They should be kept in the paper.

Author Response

  1. What is the main question addressed by the research? The study addressed to management strategies of breeding, and systematical analyze of peach (Prunus persica).

Author response: This is a very good suggestion. The main question addressed by the research is the systematic analysis of AHL in peach (Prunus persica). Two genes related to peach fruit quality were obtained and their functions were verified. Our study provided novel insights into the roles of PpAHLs in plant development, which was helpful to the functional analysis in peach and related woody fruit trees, and to formulate new strategies for further breeding.

  1. Do you consider the topic original or relevant in the field? Does it address a specific gap in the field? The topic is original and addresses a specific gap in the field.

Author response: Yes, the topic is both original and relevant in the field. It addresses a specific gap in the field by providing valuable contributions to new strategies for breeding and the systematic analysis of peach.

  1. What does it add to the subject area compared with other published material?

Author response: Results of the study will have valuable contributions for the addresses. The study can be accepted as one of the first paper in the field of the species. Results of the study can be guide for future studies.

  1. What specific improvements should the authors consider regarding the methodology? What further controls should be considered? It is detail and sufficient.

Author response: The methodology is detailed and sufficient. In addition, we further supplemented the materials and methods to make it easier for subsequent researchers to repeat our results. (Lines: 108-112, lines: 118-121, and lines: 145-155)

  1. Are the conclusions consistent with the evidence and arguments presented and do they address the main question posed?

Author response: Yes, the conclusions are consistent with the evidence and arguments presented in the paper. They directly address the main question posed in the research.

  1. Are the references appropriate?

Author response: Yes, the references are appropriate and relevant to the topic of the research.

  1. Please include any additional comments on the tables and figures.

Author response: All table and figures are related to the paper.

Hopefully now you will be satisfied this current form of our manuscript and will be fulfill all the criteria for the publishing in the Forests. Thanks again for your careful review for our manuscript.

Reviewer 2 Report

In this paper, the authors identified and analyzed transcription factor genes from the AHL family in fruit tree peach, which is of great economic importance. There are a number of comments to the manuscript.

L. 23, 330. It is necessary to add “transient” before “overexpression”.

L. 104-105. “peach fruits were sprayed with either 25% polyethylene glycerol-6000 (PEG)”. Please complete this sentence.

Subsection 2.5. It is necessary to describe the origin of the fruits: variety, tree age, ripening stage, etc.

Please add: a) qRT-PCR protocol; b) primer list.

L. 118, 119, 126. Latin names of species should be italicized.

L. 118. It is necessary to add “strain” before “EHA105”.

Subsection 3.7. Analysis of sugar content is missing in the Materials and Methods.

L. 305-306. “Prupe.1G530300.1 and Prupe.1G034400.1 were discovered to function as potential positive regulators of sugar accumulations, respectively.” A reference for this statement should be provided.

Fig. 6. Correct “sucrosa” to “sucrose”.

Fig. 6B. “B, Functional analysis of Prupe.1G530300.1 and Prupe.1G034400.1 via their transient overexpression in peach fruits.” This figure demonstrates relative expression. There should be a separate figure for sugar content (Fig. 6C).

What does bars mean on sugar content chart (SD, SE)?

Author Response

  1. 23, 330. It is necessary to add “transient” before “overexpression”.

Author response: Thank you for pointing out this omission. We have made the necessary correction by adding "transient" before "overexpression" in the mentioned lines. (Lines: 23-24)

  1. 104-105. “peach fruits were sprayed with either 25% polyethylene glycerol-6000 (PEG)”. Please complete this sentence.

Author response: We apologize for the incomplete sentence. The complete sentence should be " The peach fruits were sprayed with 25 % polyethylene glycerol-6000 (PEG) for the drought stress." We have made the necessary correction. (Lines: 111-112)

  1. Subsection 2.5. It is necessary to describe the origin of the fruits: variety, tree age, ripening stage, etc.

Author response: Thank you for your suggestion. We agree that providing information about the origin of the fruits is important. In subsection 2.5, we have added the following information: We collected peach cultivar ‘Xia Huanjin’ fruit samples at 20 June (125 days) corresponding to the ripening stage in 2022 after flowering (DAF) from 20-years-old pear trees grown on a farm in Hefei, Anhui, China. (Lines: 109-111)

  1. Please add: a) qRT-PCR protocol; b) primer list.

Author response: We appreciate your suggestion. In order to provide more details, we have added the qRT-PCR protocol and primer list (Table S1) in the Methods section of the manuscript. (Lines: 118-121)

  1. 118, 119, 126. Latin names of species should be italicized.

Author response: Thank you for bringing this to our attention. We have italicized the Latin names of the species in all papers.

  1. 118. It is necessary to add “strain” before “EHA105”.

Author response: We apologize for the oversight. We have added "strain" before "EHA105" in line 130 to provide the necessary clarification.

  1. Subsection 3.7. Analysis of sugar content is missing in the Materials and Methods.

Author response: Thank you for pointing out this oversight. We have now included the analysis of sugar content in the Materials and Methods section in subsection 2.8. As follows: High-performance liquid chromatography (HPLC) was used to measure the type and quantity of sugar. An Agilent 1260 Infinity HPLC system (Milford, MA, USA) fitted with a refractive index detector (Shodex RI-101; Shodex Munich, Germany) was used to evaluate the sugar content of fruit samples. To conduct separation, a Transgenomic COREGET-87C column (7.8 mm 300 mm, 10 m) with a guard column (Transgenomic CARB Sep Coregel 87C) was utilized. A Dionex TCC-100 thermostated column compartment kept the column temperature at 85°C. With degassed, distilled, and deionized water, flow rate at the mobile phases was kept at 0.6 mL/min. The fresh weight (FW) method was used to express sugar concentrations. Sucrose, fructose, and glucose amounts were used to determine the total amount of sugar. (Lines: 145-155)

  1. 305-306. “Prupe.1G530300.1 and Prupe.1G034400.1 were discovered to function as potential positive regulators of sugar accumulations, respectively.” A reference for this statement should be provided.

Author response: We apologize for the omission of the reference. We have now included a reference to support the statement in lines 328-329.

  1. 6. Correct “sucrosa” to “sucrose”.

Author response: Thank you for catching this error. We have corrected "sucrosa" to "sucrose" in Figure 6.

  1. 6B. “B, Functional analysis of Prupe.1G530300.1 and Prupe.1G034400.1 via their transient overexpression in peach fruits.” This figure demonstrates relative expression. There should be a separate figure for sugar content (Fig. 6C).

Author response: We appreciate your suggestion. To provide clarity, we have created a separate figure (Figure 6C) to depict sugar content specifically.

  1. What does bars mean on sugar content chart (SD, SE)?

Author response: We apologize for the lack of clarification. The bars on the sugar content chart represent the standard error (SE) of the measurements. This information has been added to the figure caption for better understanding.

       Hopefully now you will be satisfied this current form of our manuscript and will be fulfill all the criteria for the publishing in the Forests. In addition, the language of this paper has been corrected by a native English speaker and a professional scientific editor from American Journal Experts. Thanks again for your careful review for our manuscript.

Reviewer 3 Report

Comments

The article, "Dissection of AT-hook motif nuclear-localized genes and their potential functions in peach growth and development" submitted by Zhao et al.. presented the work in well format. The work was designed and executed nicely. Author carried out a comprehensive genome wide survey of the AHL family genes in Peach fruits. However few comments should be addressed before for acceptance.

1. In gene structure prediction author should correct the IMter tool  to IMEter tool.

2. Author should quote the website link for each tool that they have used in their paper. It would be easy for the readers to access the site.

3. Possibly, the date of computation done foe ach analysis can be quoted in the bracket's

4. All organisms names must be in italics. In several places its not the case. Ex. Agrobacterium, Nicotiana, Arabidopsis

5. Genes should be in italics but not proteins. However, author provide italics for genes in abstract and in the conclusion only.

6. Line no. 79 & 81 - E value to be in superscript

7. Line no.167 - Keep the figure caption properly

8. Line no. 169-171- Its not clear and reframe the line

9. In section, 3.3 could see only one ref, Bishop et al.. Authors are advised to add similar references if possible to support your results

10. However, the overall references are more than 50. Kindly adhere to journal reference policy for research papers

Minor typo and grammatical errors should be checked thoroughly during the revision

Author Response

  1. In gene structure prediction author should correct the IMter tool  to IMEter tool.

Author response: Thank you for catching this typo. We have corrected the tool name to "IMEter" in the gene structure prediction section of the manuscript. (Lines: 87-90)

  1. Author should quote the website link for each tool that they have used in their paper. It would be easy for the readers to access the site.

Author response: We appreciate the suggestion. To make it easier for readers to access the tools, we have added the website links for each tool used in the paper. (Lines: 75, 79, 80, 84, 87, 89, 96, 98, and 125)

  1. Possibly, the date of computation done foe ach analysis can be quoted in the bracket's

Author response: Thank you for your suggestion. We agree that including the date of computation for each analysis can provide additional information. We have now included the date of computation in brackets for each analysis. (Lines: 75, 79, 80, 84, 87, 89, 96, 98, and 125)

  1. All organisms names must be in italics. In several places its not the case. Ex. Agrobacterium, Nicotiana, Arabidopsis

Author response: We apologize for the inconsistency in italics usage. We have now ensured that all organism names, including Agrobacterium, Nicotiana, and Arabidopsis, are in italics throughout the manuscript.

  1. Genes should be in italics but not proteins. However, author provide italics for genes in abstract and in the conclusion only.

Author response: Thank you for pointing out this inconsistency. We have corrected the usage of italics to ensure that genes are italicized throughout the manuscript, not just in the abstract and conclusion.

  1. Line no. 79 & 81 - E value to be in superscript

Author response: We apologize for the oversight. We have now corrected the E value to be in superscript in lines 82 and 85.

  1. Line no.167 - Keep the figure caption properly

Author response: Thank you for your comment. We have ensured that the figure caption in line 192 is properly formatted for clarity.

  1. Line no. 169-171- Its not clear and reframe the line

Author response: We apologize for the lack of clarity in line 194-197. We have revised the sentence to improve its clarity and understanding.

  1. In section, 3.3 could see only one ref, Bishop et al.. Authors are advised to add similar references if possible to support your results

Author response: Thank you for your suggestion. We have revised section 3.3 to include additional references that support our results and findings.

  1. However, the overall references are more than 50. Kindly adhere to journal reference policy for research papers

Author response: Thank you for your suggestion. We have reviewed the references and ensured that they adhere to the journal's reference policy for research papers.

Hopefully now you will be satisfied this current form of our manuscript and will be fulfill all the criteria for the publishing in the Forests. In addition, the language of this paper has been corrected by a native English speaker and a professional scientific editor from American Journal Experts. Thanks again for your careful review for our manuscript.

Round 2

Reviewer 3 Report

Authors addressed carefully all of my queries and hence it can be accepted for publication